# Preparation and Micromechanics of Red Sandstone–Phosphogypsum–Cement Composite Cementitious Materials

**DOI:** 10.3390/ma16134549

**Published:** 2023-06-23

**Authors:** Chuiyuan Kong, Bin Zhou, Rongxin Guo, Feng Yan, Rui Wang, Changxi Tang

**Affiliations:** 1Yunnan Key Laboratory of Disaster Reduction in Civil Engineering, Faculty of Civil Engineering and Mechanics, Kunming University of Science and Technology, Kunming 650500, China; kongchuiyuan@stu.kust.edu.cn (C.K.); zhoubin236@126.com (B.Z.); guorx@kmust.edu.cn (R.G.); 20212210043@stu.kust.edu.cn (R.W.); tcxyywd@163.com (C.T.); 2Yunnan Transportation Development Consulting Co., Ltd., Kunming 650100, China

**Keywords:** red sandstone, phosphogypsum, flowability, mechanical strength, micromechanics, volcanic ash activity

## Abstract

Based on the physical and chemical properties of red sandstone (RS), RS is used to produce composite cementitious materials. The flowability, mechanical strength, and micromechanics of a red sandstone–cement binary cementitious material (RS-OPC) were investigated as functions of the amount of RS replacing the cement (OPC). Additionally, the feasibility of producing red sandstone–phosphogypsum–cement composite materials (RS-PG-OPC) using the phosphogypsum (PG)- enhanced volcanic ash activity of RS was investigated. The products of hydration and microstructures of RS-OPC and RS-PG-OPC were analyzed by XRD, FTIR, TG-DTG, and SEM. RS enhanced the flowability of RS-OPC relative to the unmodified cement slurry but lowered its mechanical strength, according to the experiments. When the quantity of OPC replaced was greater than 25%, the compressive strength after 28 days was substantially reduced, with a maximum reduction of 78.8% (RS-60). The microscopic mechanism of RS-OPC suggested that the active SiO_2_ in the RS can react with Ca(OH)_2_ to produce C-S-H but can only utilize small quantities of Ca(OH)_2_, confirming the low volcanic ash activity of RS. RS was responsible for dilution and filling. The incorporation of 5% PG into RS-PG-OPC slowed the hydration process compared with RS-OPC without PG but also increased the flowability and aided in the later development of the mechanical strength. This was primarily because the addition of PG provided the system with sufficient Ca^2+^ and SO_4_^2−^ to react with [Al(OH)6]^3−^ to form ettringite (AFt), therefore accelerating the dissolution of Al^3+^ in RS to generate more AFt and C-(A)-S-H gels. To some extent, this excites the volcanic ash of RS. Therefore, if there is an abundance of waste RS in the region and a lack of other auxiliary cementitious materials, a sufficient quantity of PG and a finely powdered waste RS component can be used to replace cementitious materials prepared with OPC to reduce the mining of raw OPC materials.

## 1. Introduction

In some parts of southwestern China, there is a large amount of red sandstone (RS) that dates from the Jurassic to the Neo-Proterozoic period. RS refers to red, dark red, or brown sedimentary rocks such as mudstones, murky sandstones, granular mudstones, and siltstones. They have low hardness and are easily dissolved in water to create sandstone soils because they are rich in hydrophilic clay minerals and iron oxides [1,2,3,4]. The principal chemical constituents are SiO_2_, Al_2_O_3_, CaO, and Fe_2_O_3_, and the SiO_2_ and Al_2_O_3_ content of which generally exceeds 80% [4,5,6]. In recent years, with the development of the economy and the increasing demand for transportation, the residue from excavation during the construction of roads has contained a large amount of sandstone. It is unquestionably a waste of resources and a source of environmental harm if these sandstones are dumped or thrown away.

Based on the principles of the secondary use of resources and protecting the environment, most scholars have used sandstone as the base material of pavements [7,8,9], while some scholars have also ground it into sandstone powder and used it to prepare auxiliary cementitious materials through pretreatment. He et al. evaluated the volcanic ash activity of sandstones using the lime absorption method and discovered that sandstone powder contains volcanic ash and may effectively suppress the alkali–silica reaction of active coarse sandstone aggregates in concrete, allowing it to be utilized as a partial substitute for cement (OPC) to increase concrete’s durability [5]. Some studies on the secondary use of sandstone have been carried out by Ferone and Li et al. [6,10,11,12,13,14,15]. and they have systematically studied the mechanical properties, water resistance, pore structure and hydration products of sandstone mortar geopolymers using alkali activation and heat treatment. They concluded that alkali activation and heat treatment were able to increase the reactivity of sandstone, and the oligomers prepared from the treated sandstone clay sediments had good mechanical properties. Cao et al. also pretreated sandstone by high-temperature calcination; the results showed that this not only accelerated the hydration process and improved the mechanical properties of the sandstone, but also had a beneficial filling effect in the early stages of hydration [16]. However, the ideal calcination temperature varied, and Cao et al. suggested that a calcination temperature greater than 700 °C is necessary for the optimum conversion of kaolinite to metakaolinite. At temperatures exceeding 900 °C, the amorphous metakaolinite is converted into mullite and γ-Al_2_O_3_, which reduces the sandstone’s sensitivity to volcanic ash. In addition to this, Essaidi and Selmani et al. studied the source of silica-aluminates of natural clay materials from Tunisia as low temperature synthetic consolidation materials [17,18]. Sandstone was one of the basic materials utilized by Deng and Xin et al. to create regular low-alkali silicate cement. However, they did not carry out research specifically on the process of hydration [19,20]. In summary, the current study found that sandstone has some volcanic ash activity and can be used to produce cementitious additives after pretreatment by calcination and alkali activation. It would be advantageous for both the conservation of resources and the environment if RS from Yunnan, China, could be processed into RS powder and utilized as a raw material in the manufacture of cementitious materials.

On the other hand, phosphogypsum (PG) is a solid waste produced during the wet process of producing phosphoric acid. According to statistics, for every ton of phosphoric acid produced, 4 to 6 tons of PG will be manufactured. The majority of PG is composed of calcium sulfate dihydrate (CaSO_4_·2H_2_O), the concentration of which exceeds 90%. It is acidic and contains impurities including phosphoric acid, phosphate, soluble phosphorus, fluorine, organic matter, and radioactive substances [21,22]. Many researchers have created supplemental cementitious materials by mixing PG with possible hydraulic industrial solid wastes such as blast-furnace slag [23,24], fly ash [25], steel slag [26], and red mud [27], as the major component of PG is CaSO_4_·2H_2_O. Numerous investigations have shown that PG in modest concentrations may promote volcanic ash activity in possible water-hardened industrial solid wastes. In particular, PG accelerates the dissolution of Al^3+^, Ca^2+^, and Si^4+^ in industrial residues, therefore promoting their hydrolysis to generate more C-(A)-S-H gels and AFt, therefore enhancing the strength of the composite cementitious material.

In conclusion, RS is likely to be useful in the fabrication of cementitious materials owing to its low hardness, ease of grinding, and specific volcanic ash activity. According to current research, the most effective method for preparing cementitious composite materials is high-temperature calcination or chemical reagent excitation to stimulate their volcanic ash activity. However, both high-temperature calcination and treatment with chemical reagents increase production costs and contribute to secondary contamination, so it is necessary to investigate alternative low-cost activation techniques. Because PG includes significant levels of Ca^2+^ and SO_4_^2−^, which may react chemically with activated aluminum to produce C-A-H gels and AFt in alkaline settings, it is often used to make gelling materials from industrial wastes that may have potential volcanic ash activity. However, no relevant investigations on the creation of composite cementitious materials using the PG-activated volcanic ash activity of RS have been discovered. If RS can be used as a partial substitute for OPC, the volcanic ash activity of PG-activated RS can be used to prepare composite cementitious materials, which would not only expand the secondary use of waste RS and PG and reduce the mining of raw OPC materials but would also have positive environmental and economic effects.

This study investigated the use of RS as a partial substitute for OPC to make RS-OPC and used PG to activate the volcanic ash activity of RS to show the viability of utilizing RS, PG, and OPC to produce a RS-PG-OPC that consumes less energy. This study prepared a series of samples of compound cementitious materials with varying amounts of RS-substituted OPC and 5% PG calcium, with a focus on the effects of adding RS-calcium and PG on the flowability and mechanical strength of RS-OPC and RS-PG-OPC. XRD, FTIR, TG-DTG, and SEM were used to investigate the changes in the hydration products and micromorphology, as well as the hydration of the mineral fractions of RS-OPC and RS-PG-OPC. The research results may broaden the scope of application for RS and PG wastes, reduce the cost of treating RS and PG wastes, reduce the extraction of raw OPC materials, preserve a substantial quantity of traditional resources, and lessen the environmental damage. This is extremely important for conserving resources and preserving the environment.

## 2. Materials and Methods

### 2.1. Raw Materials

The RS was obtained from Chuxiong, Yunnan Province, and processed for 30 min at 48 rpm in an SM500×500 cement test mill (Cangzhou, China). Due to the presence of Fe_2_O_3_, the RS appeared reddish-brown, as depicted in Figure 1a, and the microscopic morphology of the surface was irregular granules, as depicted in Figure 1b. The PG was gathered from Qujing, Yunnan Province. It appeared as an off-white powder with a pH of 4.17 and was acidic. Before the test, the PG was dried to a consistent weight in an oven at 50 °C and then ground for 20 min in an SM500×500 cement test mill. The OPC was an ordinary silicate cement with a content of 42.5% manufactured by Huaxin Cement Co., Ltd. (Honghe, China). The chemical composition of RS, PG, and OPC was determined using an X-ray fluorescence spectrometer (XRF) manufactured by Panalytical Ze-tium (Netherlands), and the results are shown in Table 1. Figure 2 depicts the results of determining the mineralogical composition of the RS, PG, and OPC using an X-ray diffractometer (XRD) from Rigaku SmartLab SE (Tokyo, Japan). As shown in Table 1 and Figure 2, the major mineral of PG was CaSO_4_·2H_2_O, while the main chemical composition of RS was SiO_2_, Al_2_O_3_, and CaO, with percentages of 70.80%, 9.35%, and 9.33%, respectively. Furthermore, as determined by Lee’s specific gravity bottle method and kerosene, the densities of the RS, PG, and OPC were calculated to be 2646 kg/m^3^, 2317 kg/m^3^, and 3122 kg/m^3^, respectively, according to Archimedes’ principle.

### 2.2. Mixture Proportion

We aimed to study the amount of RS replacing OPC and the effects of PG on the RS-PG-OPC. A series of mixtures was prepared with various proportions of RS replacing the OPC blended with 5% PG, as shown in Table 2. The experiment included a control (RS-0); RS with varying concentrations; and a mixture of RS, PG, and OPC. The proportion of the RS replacing OPC varied across 15%, 25%, 35%, 45%, 50%, and 60%. All the samples had the same water to binder ratio (W/B) of 0.5.

### 2.3. Preparation of the Samples

The mass of each raw material was weighed according to the mixing ratios in Table 2, and then the mixture was placed in a wet cement grout mixer and mixed at a low speed for 2 min to evenly distribute the raw materials. Water was added and mixed at a low speed for 2 min, paused for 15 s, and then mixed at a high speed for 2 min. The prepared slurry was poured into molds measuring 40 mm × 40 mm × 160 mm and cured under laboratory environmental conditions for 24 h before demolding. After demolding, the specimens were sealed with clingfilm and cured to age for the mechanical strength tests. The curing conditions were as follows: temperature, 20 ± 2 °C; relative humidity, >90%.

### 2.4. Testing Methods

#### 2.4.1. Flowability

To examine the effects of RS and PG on the flowability of RS-OPC and RS-PG-OPC, respectively, the flowability of the composite gelling materials was evaluated using the Chinese standard GB/T 8077-2012 [28]. The precise procedures entailed using 300 g of the raw materials to make the slurry as explained in Section 2.3, quickly injecting the mixed slurry into a prepared cutting cone test mold (36 mm × 60 mm × 60 mm), scraping it with a scraper, lifting the cutting cone test mold vertically, and simultaneously depressing the timer. The average value of the greatest diameter of the flowing component in two perpendicular directions was measured as the net slurry flow, and the results were the average value of three tests after the slurry had flowed on the glass plate for 30 s.

#### 2.4.2. Mechanical Strength Test

The compressive and flexural strengths were tested according to Chinese standard GB/T 17671-2021 [29] using a microcomputer-controlled electro-hydraulic cement press (YAW-300B, Jinan Shijin Group Co., Ltd., Jinan City, China) and an electric flexure testing device with a digital display (Model DKZ-6000, Wuxi Jianyi Instrument Machinery Co., Ltd., Wuxi City, China), respectively. The flexural strength test was conducted at a load rate of 50 N/s, and the test results for each ratio and each age were the average bending strength of the three prisms was determined. The loading rate in the compressive strength test was 0.5 kN/s, and the test results were characterized by the mean of six compressive strength measurements taken from a set of three prisms after flexure.

### 2.5. Characterization Methods

#### 2.5.1. X-ray Diffractometry (XRD)

Before the characterization of the hydration products and microstructures with XRD, TG-DTG, FTIR, and SEM, it was necessary to carry out the solvent exchange in the samples to prevent further hydration. After the compressive strength test was completed, the internal fragments were extracted and immersed in anhydrous ethanol for 48 h. The resulting specimens were vacuum-dried at 40 °C for 48 h before being pulverized into powder for the XRD, TG-DTG, and FTIR analyses. All tested samples were subjected to XRD analysis using a Rigaku SmartLab SE (Japan) X-ray diffractometer with a Cu K X-ray source at a voltage of 40 kV and a current of 40 mA to determine the mineralogical composition of RS-OPC and RS-PG-OPC. The scanning range was 5–60°, with a 2°/min scanning speed.

#### 2.5.2. Fourier Transform Infrared Spectroscopy (FTIR)

Before testing, each powdered sample was mixed with potassium bromide at a ratio of 1:100 (KBr: spectral grade, purity: ≥99.5%, produced by Shanghai Aladdin Biochemical Technology Co., Ltd., Shanghai, China), compressed into tablets (Tablet press model: HY-15, produced by Tianjin Tianguang Optical Instruments Co., Ltd., Tianjin, China; die hole size: φ = 13 mm; pressure: 8 MPa), and analyzed using a Thermo Scientific iN10 (Waltham, MA, USA) infrared spectrometer in the mid-infrared region of 400–4000 cm^−1^ to characterize the products of hydration. The spectrometer featured a 4 cm^−1^ resolution and 32 scans.

#### 2.5.3. Thermal Analysis (TG-DTG)

The samples were analyzed by TG-DTG to examine the effects of the amount of RS replacing OPC and the addition of PG on the hydration products of RS-OPC and RS-PG-OPC. The samples were heated from 30 °C to 900 °C at a heating rate of 10 °C/min in an N_2_ atmosphere at a flow rate of 50 mL/min using a Rigaku TG/DTA8122 (Japan) calorimeter.

#### 2.5.4. Scanning Electron Microscopy (SEM)

Using a Czech TESCAN MIRA LMS scanning electron microscope, the microscopic morphological characteristics of the hydration products of the composite gelling materials were observed. The accelerating voltage was 15 kV. The samples were coated with gold using ion sputtering before being viewed.

The laser particle size method was used to characterize the fineness of the RS, PG, and OPC, and the Malvern Mastersizer 2000 (London, UK) laser particle size analyzer was used to ascertain the particle size distribution of the three raw materials. The results are shown in Figure 3. Of all the raw materials, RS had the finest average particle size of D_50_ = 14.426 μm, while PG had the coarsest particle size of D_50_ = 36.977 μm.

Figure 4 shows the FTIR spectra of RS and PG. The high-frequency (4000–1600 cm^−1^) spectral bands of PG were related to the asymmetric and symmetric stretching vibrations of the O-H bond in water [30]. The vibrational absorption bands of the S-O bond were observed at 1115 cm^−1^, 668 cm^−1^, and 601 cm^−1^, and the 468 cm^−1^ bending vibrations of SiO_4_ were associated with the quartz in PG [30]. In addition, the band at 839 cm^−1^ was associated with the eutectic phosphorus in PG [31]. For RS, the asymmetric stretching band of -Si-O-H was observed at 3424 cm^−1^ [30]. Furthermore, the Si-O stretching band at 1084 cm^−1^ was associated with feldspar, whereas the Si-O vibrational band at 1035 cm^−1^ was associated with quartz [5]. The two absorption bands at 812 cm^−1^ and 465 cm^−1^ corresponded, respectively, to the Al-O and Si-O bonds [32].

## 3. Results and Discussion

### 3.1. Flowability

As shown in Figure 5, the flowability of RS-OPC increased as the replacement rate of RS increased compared with the original cement paste (RS-0). In particular, the flowability increased considerably when the rate of RS reached 25% (RS-25). Nonetheless, when the rate of RS exceeded 35% (RS-35), the growth rate of flowability declined substantially with an increase in the proportion of RS. For example, the flowability was 150 mm for RS-0, 154 mm for RS-25, 170 mm for RS-35, and 174 mm for RS-60, increasing by 2.7%, 13.3%, and 16.0%, respectively. This suggested that the addition of RS can enhance the flowability of RS-OPC. This is due to the filling effect of RS. The smaller size of RS powder fills the space between OPC particles, optimizes the particle accumulation and particle size distribution of RS-OPC, reduces the porosity of RS-OPC, decreases the pore size, and improves the compactness of RS-OPC. Therefore, the filling effect of RS can reduce the water requirement of RS-OPC under a certain amount of admixture [33]. Thus, the addition of RS improves the flowability of RS-OPC at the same W/B. In addition, the presence of PG increased the flowability of RS-PG-OPC, with the sample containing 5% PG (RPO-5) exhibiting a 9.3% increase in flowability relative to RS-25. This was because the average particle size of PG is significantly larger than that of RS and OPC. The addition of PG decreased the water demand of RS-PG-OPC, therefore increasing its flowability [34].

### 3.2. Effect of Different RS Replacement Amounts on the Mechanical Strength of RS-OPC

The effects of different replacement amounts of RS on the mechanical strength of RS-OPC are shown in Figure 6. The results showed that the mechanical strength of RS-OPC decreased with increasing amounts of RS compared with RS-0, similar to He et al.’s conclusion [5]. However, it should be noted that the reduction was not proportional. When the amount of RS was 25% (RS-25), the compressive strength at 28 days was reduced by 25.9% compared with zero RS (RS-0). When the proportion of RS exceeded 25%, the compressive strength of the specimens decreased dramatically, with a maximal decrease of 78.8% at 28 days (RS-60). This indicated that excessive RS replacement can result in RS-OPC with a severe loss of strength. In agreement with the micromechanical analysis in Section 3.3, the amount of RS replacing OPC will reduce the number of cementitious materials, which will reduce the amount of hydration products per unit of volume in RS-OPC. RS acts as a diluent [33], contributing less to the development of the strength of RS-OPC due to the low volcanic ash activity of the as-received RS and the small percentage of its participation in the reaction. Therefore, the maximum replacement of OPC by RS should not exceed 25% by weight.

In general, a significant linear relationship exists between compressive strength and flexural strength [35]. It can be seen in Figure 6c that the compressive strength and flexural strength had a significant linear correlation (R^2^ = 0.95924). This suggests that the equation in the figure appropriately represented the relationship between compressive and flexural strength.

### 3.3. Microscopic Mechanisms of RS-OPC with Different Amounts of RS 

#### 3.3.1. XRD Analysis

The XRD findings of RS-0, RS-25, and RS-60 after 1 day, 3 days, and 28 days of hydration are shown in Figure 7. As the diagram shows, the main mineral phases in the hardened slurry of the samples were ettringite (AFt), calcium hydroxide (Ca(OH)_2_), hydrated (aluminum) calcium silicate gel (C-(A)-S-H), calcium silicate (C_2_S/C_3_S), quartz (SiO_2_), calcite (CaCO_3_), and feldspar (NaAlSi_3_O_8_). AFt, C-(A)-S-H gels, and Ca(OH)_2_ are among the hydration products of the chemical process, while the raw material contained C_2_S/C_3_S, SiO_2_, CaCO_3_, and NaAlSi_3_O_8_. Additionally, throughout the storage period, CaCO_3_ was connected to the carbonation of the samples’ hydration products. There are no significant diffraction peaks in the XRD spectrum because C-(A)-S-H gels are amorphous non-crystalline and correspond to amorphous dispersion peaks in the 20°–40° range in the XRD spectrum [36].

For the RS-containing samples (RS-25, RS-60), the strength of the SiO_2_ and Ca(OH)_2_ diffraction peaks reduced to varying degrees with hydration age compared with RS-0. In particular, the Ca(OH)_2_ in the hydration product of OPC underwent a secondary hydration reaction with the active SiO_2_ in the RS, as the intensity of the SiO_2_ diffraction peak was significantly less after 3 days of hydration than after 1 day. However, a significant quantity of Ca(OH)_2_ remained after RS-60 had been hydrated for 28 days, showing that the amount of volcanically active SiO_2_ in RS had been constrained. Moreover, as the proportion of RS increased, the intensity of the AFt diffraction peaks in the cured slurry of the samples decreased to varying degrees relative to RS-0. This indicated that RS was unable to stimulate the formation of AFt in the gelling system. To summarize, the active SiO_2_ in the RS powder could consume Ca(OH)_2_ in the hydration products of OPC, but its ability to participate in the chemical reaction was limited, indicating that the volcanic ash activity of the as-received RS was low and only a small amount of active SiO_2_ was involved in the hydration reaction, and the addition of RS reduced the number of hydration products in the system, lowering the overall strength of the RS-containing samples.

#### 3.3.2. FTIR Analysis

FTIR spectroscopy can be used as a supplement to XRD to characterize the binding properties of the crystalline and amorphous phases of a substance. It was also used to determine the chemical concentration and gel polymerization through the strength and breadth of the absorption bands [30]. Figure 8 depicts the FTIR results of the three samples (RS-0, RS-25, and RS-60) after 28 days of curing. The FTIR spectra of the various samples were extremely similar, and the absorption regions in the spectra were essentially identical, but the absorption varied. The absorption peak at 3640 cm^−1^ was caused by the Ca-OH in Ca(OH)_2_ [37,38]. The absorption peak at 3440 cm^−1^ was consistent with the stretching vibration of the [Al(OH)6]^3−^ (Al-OH) functional group in AFt [37,39,40]. The bending vibration of H-O-H in the hydration products corresponded to the absorption peak at 1636 cm^−1^ [41]. The characteristic absorption peak at 1430 cm^−1^ was an asymmetric stretching vibration absorption peak of O-C-O [42], and the characteristic absorption peak observed here was related to the CO_3_^2−^ ion vibrations of calcite in the carbonation of the unprocessed RS and C-S-H [36] during the hardening of the sample [43]. The asymmetric stretching vibration of Si-O-T (T = Al or Si) was linked to the distinctive absorption peak at 995 cm^−1^ [44,45]. This was equivalent to the C-(A)-S-H gels of the hydration products. The 465 cm^−1^ absorption peak corresponded to the Si-O bond, which was caused by the symmetric variable vibration angle of Si-O in the raw RS material [46], which was associated with quartz in the raw material and was also a characteristic peak of C-S-H gels [44]. Additionally, the absorption band at 528 cm^−1^ was linked to the crystalline Si-O-Al bond [16].

As shown in Figure 8, the sharpness of the absorption peak located at 3640 cm^−1^ decreases progressively with an increase in RS, which is correlated with the decrease in the concentration of Ca(OH)_2_. With increasing proportions of RS, the intensity of the absorption maxima at 3440 cm^−1^ and 1636 cm^−1^ decreased to varying degrees, which was related to the decrease in the generation of AFt, which was one of the primary causes of the decrease in the strength of the cementitious composite. Although the addition of RS did not significantly reduce the amount of C-(A)-S-H gel because of the capacity of the partially active SiO_2_ to consume a certain amount of Ca(OH)_2_ and produce C-S-H gels, the intensity of the absorption peak of RS-60 at 995 cm^−1^ was not significantly different from that of RS-0 and RS-25. In conclusion, the addition of RS reduced the quantity of AFt in the system, but the modest amount of active SiO_2_ may have encouraged the secondary hydration process of Ca(OH)_2_ to generate C-S-H gels. The results of the FTIR analysis concurred with those of the XRD analysis, further validating the minimal activity of RS.

#### 3.3.3. TG-DTG Analysis

Figure 9 displays the mass loss of three samples (RS-0, RS-25, and RS-60) after 28 days of hydration at various temperatures. Figure 9a shows three separate heat absorption peaks. It is generally accepted that the dehydration of C-(A)-S-H gels and AFt is the primary cause of mass loss in the temperature range of 40 °C to 200 °C [15,46]. The thermal decomposition of Ca(OH)_2_ is what causes the considerable peak that is present between 360 °C and 470 °C [45]. The thermal breakdown of CaCO_3_ is mostly responsible for the mass loss in the 600–750 °C range [47].

Figure 9a depicts the TG-DTG curves of the samples between 30 °C and 900 °C, and Figure 9b depicts the rate of mass loss of the samples in distinct temperature ranges. The corresponding amount of hydration products can be inferred from the mass loss of the samples in different temperature ranges [40]. The three samples’ rates of mass loss in the range of 40–200 °C were 11.37%, 8.50%, and 4.87%, respectively for RS-0, RS-25, and RS-60. The content of C-(A)-S-H gels and AFt in the samples showed a decreasing trend as the proportion of RS increased, with the rate of mass loss of RS-25 being 74.76% that of RS-0, and that of RS-60 being only 42.83% that of RS-0. It can be hypothesized that excessive supplementation with RS reduces the amount of C-(A)-S-H gel and AFt in RS-OPC, both of which are directly related to the development of mechanical strength. This is the primary cause of the decrease in mechanical strength. The three samples’ relative rates of mass loss in the range of 360–470 °C were 4.22%, 3.00%, and 2.62%, respectively, for RS-0, RS-25, and RS-60. This demonstrated that the Ca(OH)_2_ level of the RS samples was lower than that of RS-0. One possible explanation is that the addition of RS decreased the relative quantity of CaO in the system, resulting in a reduction in Ca(OH)_2_ in the hydration products. Another possibility is that the Ca(OH)_2_ in the RS was consumed further by a subsequent hydration reaction with active SiO_2_ to generate C-S-H gels. Furthermore, it is interesting that the difference in the rate of mass loss of Ca(OH)_2_ between RS-25 and RS-60 was not statistically significant, suggesting that there was only a finite amount of reactive SiO_2_ in RS that could react with the Ca(OH)_2_ to form hydrates. This further supported the low activity of RS. The thermal decomposition of CaCO_3_ brought on by the carbonization of C-S-H gels during the hydration of the samples, on the one hand, and the increase in the calcite in RS, on the other, caused the mass loss in the range of 600–750 °C to increase with the increase in the RS content in the raw material. The results of the TG-DTG analysis were consistent with the outcomes of the mechanical strength tests, XRD, and FTIR. The addition of too much RS will greatly reduce the amount of hydration products in RS-OPC, which will result in a decrease in the strength of RS-OPC.

#### 3.3.4. SEM Analysis

Figure 10 depicts the results of the SEM analysis of the morphology of the hardened slurry of three samples (RS-0, RS-25, and RS-60) to reveal the microstructure of the RS-OPC binary gelling material after 28 days of hydration. The microstructure of the hardened slurry changed substantially as the amount of RS increased. Compared with RS-0, the RS-25 and RS-60 samples had many pores, larger pores than the control group, and a looser overall structure. In addition, the density of long needle-like or short columnar AFt [32] is significantly weakened. However, the number of fibrous amorphous C-(A)-S-H gels [32,48] increased in RS-25 compared with RS-0, whereas this was not observed in RS-60, indicating that the active SiO_2_ in RS could react with Ca(OH)_2_ by hydration to form C-S-H gels, but the amount that can participate in the reaction is limited, further confirming the low activity of RS. In addition, it was seen that when the rate of RS increased, a significant quantity of unreacted RS particles (Figure 1b) manifested itself in the sample, coating the surface of the hydration products and filling the pores. In conclusion, a reasonable quantity of RS may encourage the creation of C-S-H gels, whereas an excessive amount of RS mostly fills gaps and hinders the growth of the cured slurry’s strength.

### 3.4. Effect of PG on the RS-PG-OPC’s Mechanical Strength

Figure 11 depicts the impact of PG on the mechanical strength of RS-PG-OPC. The compressive strength of the RPO-5 samples at 1 day, 3 days, and 7 days decreased by 66%, 54%, and 25%, respectively, while the compressive strength at 28 days was comparable with that of RS-25; the flexural strength followed a similar trajectory. This indicated that the addition of PG did not diminish the long-term potency of RS-PG-OPC. The early mechanical strength of PG-containing samples was substantially lower because impurities in the PG retarded the hydration of the cement [49], resulting in a reduced content of early hydration products produced by RS-PG-OPC, therefore lowering its early strength. To further investigate the specific effects of PG on the long-term mechanical strength of the RS-PG-OPC, a detailed micro-mechanistic analysis is presented below.

### 3.5. Effect of PG on the RS-PG-OPC’s Microscopic Mechanisms

#### 3.5.1. XRD Analysis

Figure 12 depicts the numerous mineral phases, including AFt, CaSO_4_·2H_2_O, Ca(OH)_2_, C_2_S/C_3_S, SiO_2_, and CaCO_3_, after 1 day, 3 days, and 28 days of hydration for RPO-5 and RS-25. The figure makes it evident that although the strength of the diffraction peak of AFt progressively rose and CaSO_4_·2H_2_O was present throughout the curing time, the intensity of the diffraction peak of CaSO_4_·2H_2_O in RPO-5 gradually declined as the curing time extended. This indicated that PG was not completely consumed during the formation of AFt and there was still a surplus, even though PG was present throughout the entire hydration process. According to Ma et al., the SO_4_^2−^ produced by PG may combine with the hexa-ligand [Al(OH)6]^3−^ to generate AFt, and the system’s reduced concentration of Al^3+^ can encourage the dissolution of reactive Al_2_O_3_ and increase the production of C-A-H and AFt [48], which can increase the RS-PG-OPC’s strength. Figure 12 illustrates that after 28 days of hydration, the intensity of the diffraction peak of AFt in RPO-5 was greater than that in RS-25, which was consistent with the active Al_2_O_3_ in the RS playing a role in the hydration process. Moreover, after 28 days of hydration, the intensity of the diffraction peaks of SiO_2_ and Ca(OH)_2_ in the XRD patterns of RPO-5 also decreased to varying degrees in comparison with RS-25, suggesting that the addition of PG may have encouraged the dissolution of active Al_2_O_3_ in RS and consumed some Ca(OH)_2_, which improved the later intensity of RS-PG-OPC. In conclusion, although adding PG slowed down the hydration process, it was able to speed up the hydrolysis of RS, and the right quantity of PG benefited the subsequent increase in the strength of the RS-PG-OPC.

#### 3.5.2. FTIR Analysis

Figure 13 shows the FTIR findings for RPO-5 for various hydration times and for RS-25 after 28 days of hydration. Except for the absorption peak at 1115 cm^−1^, which was associated with SO_4_^2−^ in PG [50,51], the corresponding chemical bonds or functional groups at other wave numbers were identical to those described in Section 3.3.2. The existence of SO_4_^2−^ absorption peaks in RPO-5 throughout the hydration process indicated that PG did not entirely contribute to the formation of AFt. Consistent with the analysis of the XRD patterns, after 28 days of hydration, the absorption peak of RPO-5 near 3440 cm^−1^ was sharper than that of RS-25, indicating that the amount of Al-OH in RPO-5 was greater than that in RS-25 and that this group was associated with the production of AFt. As can be seen from the XRD patterns, RPO-5 contained more AFt than RS-25 did, and the strength of the diffraction peak of AFt was likewise higher than that of RS-25. The strength of RPO-5’s absorption peak at 1636 cm^−1^ was also stronger than that of RS-25. This means that after 28 days of hydration, RPO-5 had more bound water than RS-25. In addition, the intensity of the absorption peak at 995 cm^−1^ increased gradually with the hydration time, becoming especially pronounced from 3 days to 28 days, whereas the intensity of the absorption peak at 1115 cm^−1^ decreased, indicating that the addition of PG facilitated the hydrolysis of RS to produce more C-(A)-S-H gels. Intriguingly, even though the amount of hydration products (C-(A)-S-H and AFt) in RPO-5 was greater than that in RS-25, the mechanical strength was comparable because the PG that was not involved in the reaction acted as an aggregate filler, whereas the strength of the PG crystals was extremely low [48] and contributed little to the strength of RS-PG-OPC. Therefore, adequate quantities of PG can enhance the volcanic ash activity of RS and the later intensity of RS-PG-OPC. The FTIR results are equivalent to the findings of the XRD analysis.

#### 3.5.3. TG-DTG Analysis

Figure 14 shows the mass loss of RPO-5 after 1 day, 3 days, and 28 days of hydration at various temperatures. Except for the heat-absorbing peak at 130 °C, which was brought on by the dehydration of CaSO_4_·2H_2_O [52], the positions of the three heat-absorbing peaks and the types of hydration product were identical to those described in Section 3.3.3. It can be observed from the TG-DTG curves in Figure 14a that the peak of CaSO_4_·2H_2_O gradually decreased with the increase in the curing time but did not disappear completely, which was consistent with the XRD and FTIR results. The rate of mass loss of the samples at various temperatures and times is shown in Figure 14b. The rate of mass loss of RPO-5 at 40–200 °C was 10.88% after 28 days of hydration, which was greater than that of RS-25, which was 8.50%. This suggests that the addition of PG stimulated the formation of AFt and C-(A)-S-H gels, consistent with the XRD and FTIR measurements of RPO-5 after 28 days. The addition of PG supplies sufficient Ca^2+^ and SO_4_^2−^ for the formation of AFt, and, within a reasonable range, the higher the SO_4_^2−^ content, the greater the production of calcium meteorites. In addition, SO_4_^2−^ was able to alter the system’s charge distribution [52], which accelerated the dissolution of Al^3+^ in RS to produce more AFt and C-A-H gels, therefore enhancing the mechanical strength of RS-PG-OPC. In general, the greater the amount of water chemically combined, the greater the degree of hydration of the hardened plasma, and the greater the amount of hydrated product produced, the greater the mechanical strength of the cementitious material [53]. Despite this, the total rate of mass loss of RPO-5 was greater than that of RS-25 after 28 days of hydration, while the mechanical strength of RPO-5 at 28 days was similar to that of RS-25. This disparity may be attributable to the water loss of the unreacted PG in RPO-5.

#### 3.5.4. SEM Analysis

Figure 15 shows the microscopic morphology of RPO-5 after 1 day, 3 days, and 28 days of curing, and RS-25 after 28 days of curing. The amount and shape of the C-(A)-S-H gels, AFt, and RS substantially altered with an increased curing time. These substances were detected at various timepoints. After 1 day of curing, only a few C-(A)-S-H gels and AFt had formed, and the microstructure was primarily composed of unreacted RS and PG crystals with a sheetlike structure [48]. After 3 days of curing, the system began to create increasingly fibrous C-(A)-S-H gels and needle-like AFt; however, at this stage, the AFt mostly was in the form of tiny needles with subpar mechanical qualities. In addition, RS was considerably lower than at 1 day. As the hydration reaction progressed, many short-column AFt particles were observed in the microstructure, which were frequently interwoven and filled with gaps, forming a relatively dense microstructure with C-(A)-S-H gel packages. Compared with the microstructure of RS-25 after 28 days of cured (Figure 15d), the amount of AFt in RPO-5 after 28 days of cured was substantially higher (Figure 15c). This indicated that the addition of PG promoted the formation of AFt in the RS-PG-OPC slurry and enhanced the late strength of RS-PG-OPC, which was consistent with the XRD, FTIR, and TG-DTG results. It is well known that the greater the number of hydration products, the greater the strength of the cementitious material [54], and the strength of RPO-5 was comparable with that of RS-25 because although the AFt content in RPO-5 was relatively high, its overall compactness was not as good as that of RS-25, and the PG that was not involved in the hydration reaction acted as a filler and as an aggregate, and contributed less to the strength. As a result, the right quantity of PG may encourage the synthesis of AFt and C-(A)-S-H gels in the system and have a good impact on the volcanic ash activity of RS.

## 4. Conclusions

To expand the use of RS in gelling materials because of its physical and chemical properties, this study prepared RS-OPC binary gelling materials by substituting the OPC content with varying mass fractions of RS. On this basis, an RS-PG-OPC ternary composite cementitious material was prepared using 5% PG to excite the volcanic ash activity of RS, and the changes in the flowability, mechanical strength, hydration products, and microstructure were analyzed. The conclusions are as follows:RS could increase the flowability of RS-OPC but diminished its mechanical strength. When the proportion of RS exceeded 25%, the RS-OPC’s mechanical strength decreased considerably. On the other hand, 5% of PG improved the flowability and promoted the development of the mechanical strength of the RS-PG-OPC at a later stage.Ca(OH)_2_, the hydration product of OPC, was capable of secondary hydration reactions with active SiO_2_ in RS to form C-S-H gels, but the amount of Ca(OH)_2_ that could be consumed was extremely limited, indicating that RS had some volcanic ash activity, although the level of activity was low, and that RS primarily demonstrated dilution and filling effects.Compared with RS-OPC without PG doping, the addition of PG delayed the hydration reaction of RS-PG-OPC, but it facilitated the ensuing development of strength. Specifically, the addition of PG introduced enough Ca^2+^ and SO_4_^2−^ into the system, which could react with [Al(OH)6]^3−^ to form Aft, and reduced the concentration of Al^3+^ in the system, therefore promoting the dissolution of RS and stimulating the volcanic ash activity of RS to produce more C-(A)-S-H gels and AFt, and consuming part of the Ca(OH)_2_, which improved the late strength of the cured slurry.If there is a large amount of waste RS in the area and there is a lack of other cementitious additives, a composite cementitious material can be prepared using an appropriate amount of PG to activate the RS powder that is ground finely enough to partially replace the OPC. This would reduce the extraction of raw materials for OPC while expanding the secondary use of waste RS and PG, which would be advantageous in terms of raw materials, economy, and ecology. In addition, it is necessary to continue to explore other environment-friendly and cost-effective methods of RS activation in future research in this area. In addition, the feasibility of specific applications of this cement, such as using this material as a binder for bedding concrete, sidewalk and road base materials with relatively low strength requirements.

## Figures and Tables

**Figure 1 materials-16-04549-f001:**
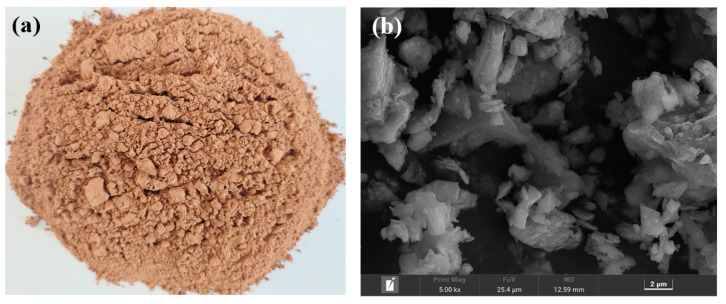
Appearance of the red sandstone granules (**a**) and SEM image (**b**).

**Figure 2 materials-16-04549-f002:**
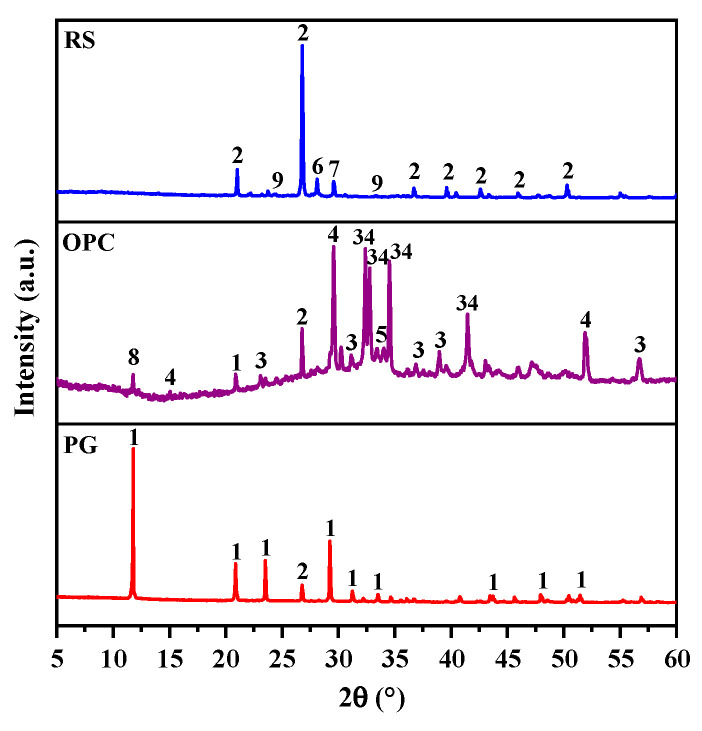
XRD spectra of the raw materials: 1, calcium sulfate dihydrate; 2, quartz; 3, dicalcium silicate; 4, tricalcium silicate; 5, tricalcium aluminate; 6, feldspar; 7, calcite; 8, gypsum; 9, hematite.

**Figure 3 materials-16-04549-f003:**
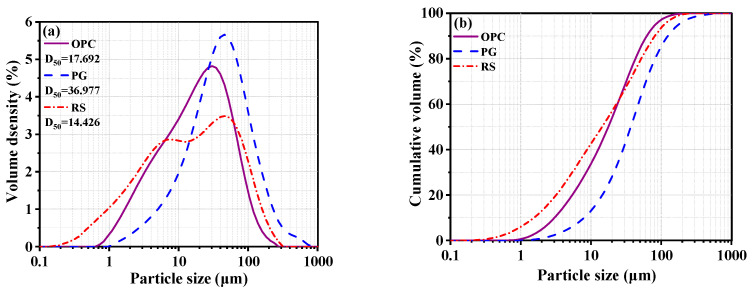
Particle size distribution of raw materials ((**a**): Volume density; (**b**): Cumulative volume).

**Figure 4 materials-16-04549-f004:**
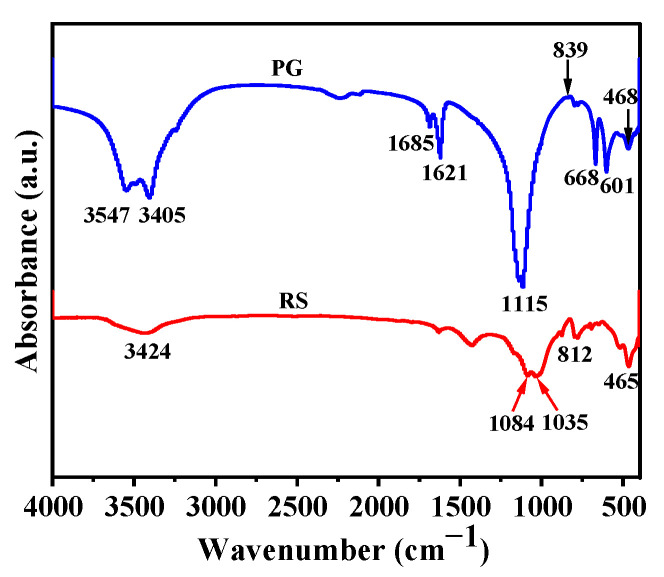
FTIR spectra of PG and RS.

**Figure 5 materials-16-04549-f005:**
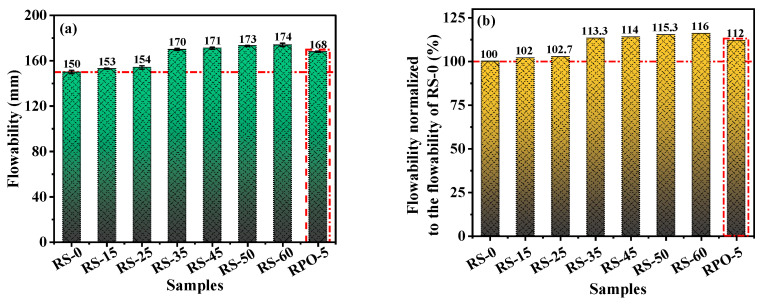
The samples’ flowability (**a**) and flowability normalized to the flowability of RS-0 (**b**).

**Figure 6 materials-16-04549-f006:**
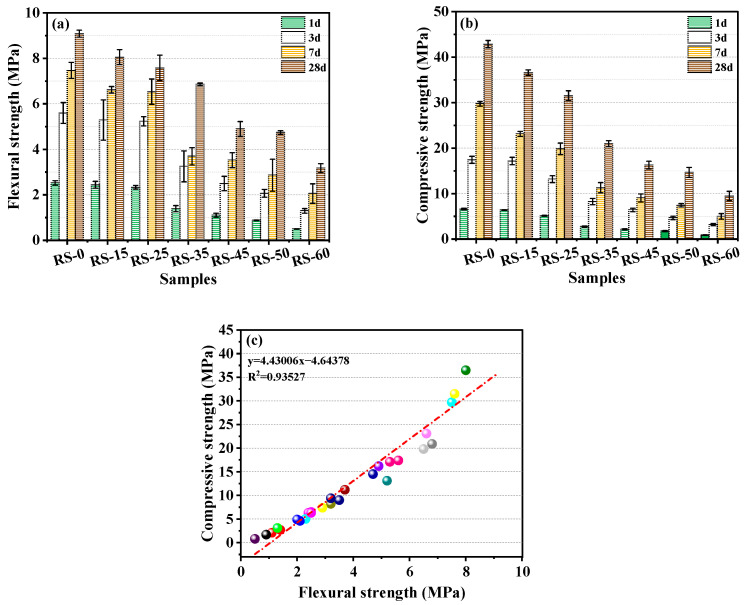
The influence of various RS replacement rates on the flexural strength (**a**) and compressive strength (**b**) and the relationship between the compressive strength and flexural strength (**c**).

**Figure 7 materials-16-04549-f007:**
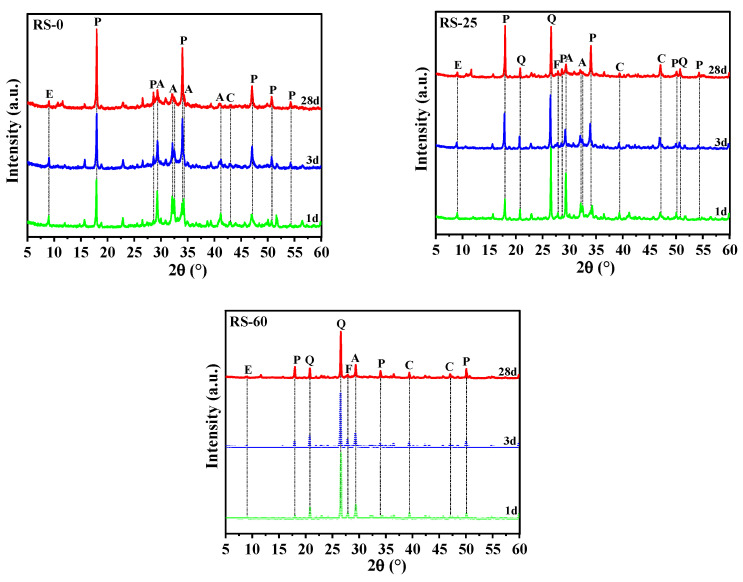
XRD patterns of various cured pastes. E, ettringite; P, portlandite; Q, quartz; A, calcium silicate; C, calcite; F, feldspar.

**Figure 8 materials-16-04549-f008:**
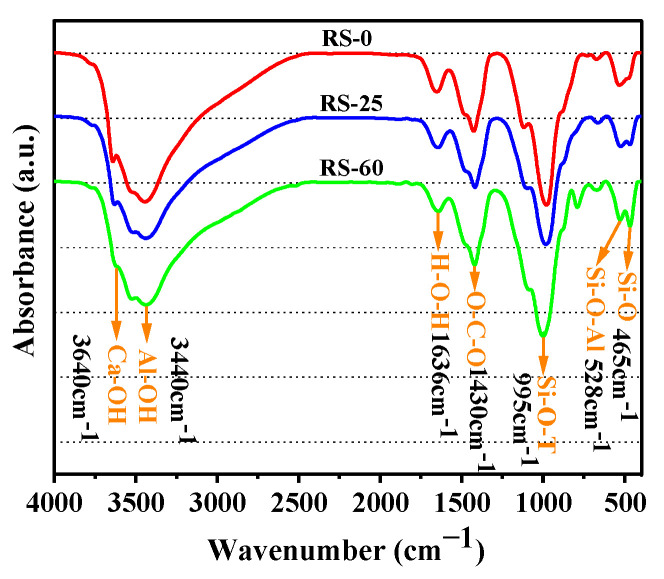
FTIR spectra of various samples cured for 28 days.

**Figure 9 materials-16-04549-f009:**
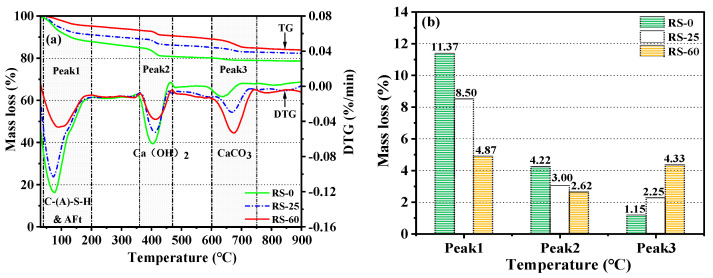
TG-DTG after 28 days of curation at various RS substitution rates (**a**) and mass loss at various temperature ranges (**b**).

**Figure 10 materials-16-04549-f010:**
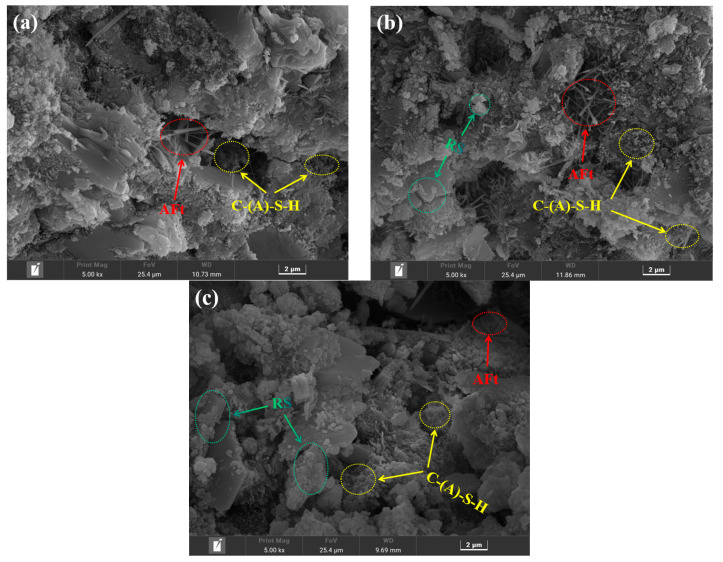
SEM images of RS-0 (**a**), RS-25 (**b**), and RS-60 (**c**) after 28 days of curation.

**Figure 11 materials-16-04549-f011:**
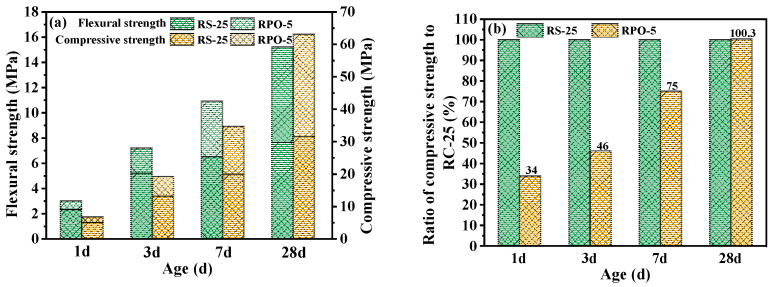
Mechanical properties of RS-PG-containing samples (**a**) and the compressive strength relative to RS-25 (**b**).

**Figure 12 materials-16-04549-f012:**
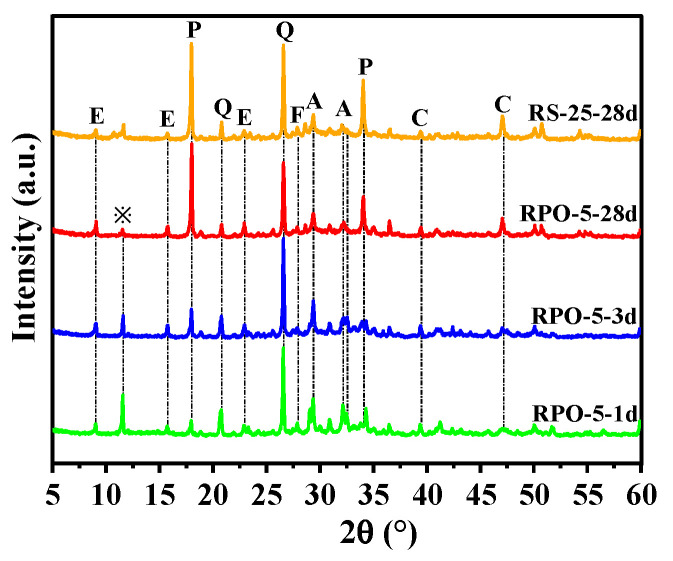
XRD patterns of RPO-5 after 1 day, 3 days, and 28 days of curation and RS-25 after 28 days of curation. E, ettringite; ※, calcium sulfate dihydrate; P, portlandite; Q, quartz; A, calcium silicate; C, calcite; F, feldspar.

**Figure 13 materials-16-04549-f013:**
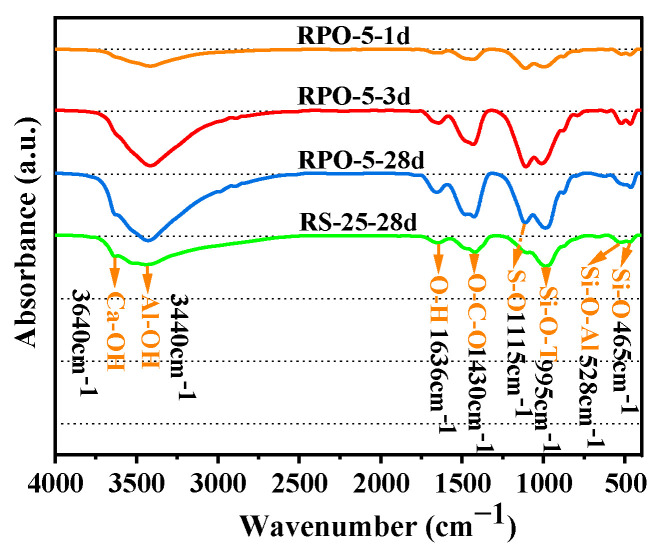
FTIR of RPO-5 cured for 1 day, 3 days, and 28 days, and RS-25 after 28 days of curation.

**Figure 14 materials-16-04549-f014:**
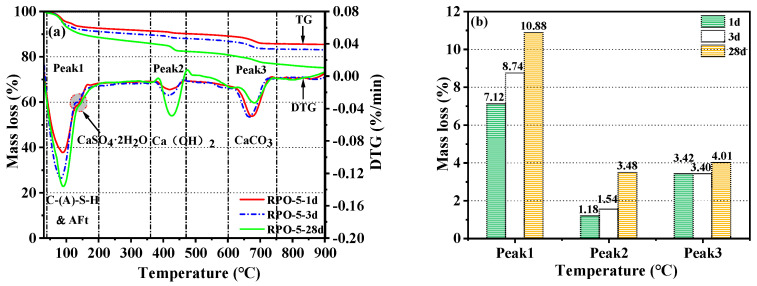
TG-DTG of RPO-5 at various ages (**a**) and mass loss at different temperatures (**b**).

**Figure 15 materials-16-04549-f015:**
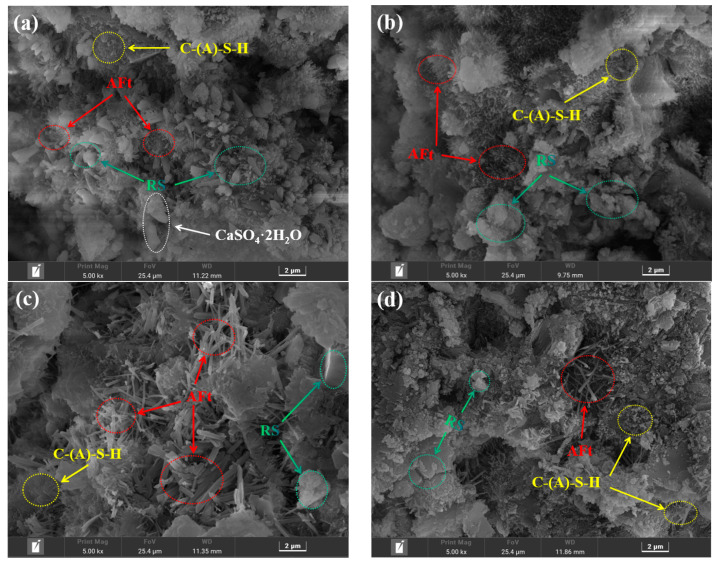
SEM images of RPO-5 cured for 1 day (**a**), 3 days (**b**), and 28 days (**c**), and of RS-25 after 28 days of curation (**d**).

**Table 1 materials-16-04549-t001:** Chemical composition of the raw materials (wt.%).

Materials	CaO	SiO_2_	Al_2_O_3_	SO_3_	Fe_2_O_3_	MgO	Na_2_O	K_2_O	P_2_O_5_	Other
RS	9.33	70.80	9.35	0.03	3.93	2.07	1.63	1.78	0.69	0.39
PG	37.88	10.54	0.45	49.97	0.18	-	-	0.13	0.12	0.73
OPC	61.13	19.88	5.49	4.23	3.36	3.11	0.29	0.80	0.18	1.53

**Table 2 materials-16-04549-t002:** Proportions of the mixed specimens (wt.%).

Samples	RS/%	OPC/%	PG/%	W/B
RS-0	0	100	0	0.5
RS-15	15	85	0	0.5
RS-25	25	75	0	0.5
RS-35	35	65	0	0.5
RS-45	45	55	0	0.5
RS-50	50	50	0	0.5
RS-60	60	40	0	0.5
RPO-5	22.5	72.5	5	0.5

## Data Availability

The data presented in this study are available from the corresponding author upon reasonable request.

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
