# Peer review of "Preparation and Micromechanics of Red Sandstone–Phosphogypsum–Cement Composite Cementitious Materials"

_materials, 2023, doi:10.3390/ma16134549_

Round 1
Reviewer 1 Report
The paper is interesting and well written. Some considerations are raised by the reviewer to improve the clarity and the quality of the paper that can be considered for publication after some minor changes. The comments are listed below:
- The authors should clarify better how many samples were tested for each configuration and for each age-of-concrete;
- The concrete specimens with no replacement of RS provided around 43 MPa of compressive strength after 28 days and the reduction in strength was around 80% in the case of RS 60. The authors should spend some consideration about the role and the influence of the concrete strength level i.e. in the case of concrete RS-0 with higher and/or lower compressive strength. Do the authors expect the same trend of reduction in the case of adding the same percentage of RS? Please discuss;
- It seems that there is second order law relationship between the compressive strength and the flexural strength (Fig. 6). Please motivate or change the fitting law;
- The authors should spend some considerations about the field of use of this material i.e. bearing structures, foundation, pavements, roads or other
Author Response
Dear Reviewers:
We gratefully thank your constructive comments and helpful suggestions, which has significantly raised the quality of the manuscript and has enabled us to improve the manuscript. Each of your suggested revisions and comments was thoroughly considered and modified in the Revised Manuscript. In this file, the authors will answer the comments item by item.
Thank you again for your time and profound comments on our manuscript.
Best regards,
Chui-yuan Kong

Reviewer 2 Report
Comments on the manuscript titled: Preparation and micromechanics of red sandstone–phosphogypsum–cement composite cementitious materials by Chuiyuan Kong, Bin Zhou, Rongxin Guo, Feng Yan, Rui Wang and Changxi Tang
The subject of the paper, titled 'Preparation and micromechanics of red sand- stone-phosphogypsum-cement composite cementitious materials', is the use of red sand as a reactive additive to Portland cement. In this paper, the authors focus on the experimental and statistical determination of the effect of varying red sand content in a Portland cement mixture and its influence on the phase composition and selected physicochemical properties. I find this work interesting, however, well covered by the world literature. I find the results obtained in the present work practical and technologically useful. The research is planned and executed quite carefully. Please find some comments as below:
- Please label all diffraction peaks in Figure 2. A precise analysis of the phase composition of the initial samples forms the basis for interpreting the interrelationships in the hydration processes in further stages of the work. Furthermore, in future work, please also be encouraged to perform a quantitative analysis.
- Did the authors of the paper perform a roasting loss analysis of the starting raw materials? Performing such an analysis for the RS sample is crucial. In the diffractogram of the RS sample, a diffraction peak originating from the calcium carbonate phase is visible, but no account has been taken of how the iron oxide phase is bound in the raw material. In the mineral composition, the iron oxide is probably present as a hydroxide or some Fe ion may be bound in the scale. The authors of this paper will not consider the potential reactivity of iron compounds in RS in mixtures with OCPs
- Planning the preparation of OCP and RS mixtures should begin with testing the activity of RS in water and lime water (pH = 11) then the likely behaviour of the RS additive in the alkaline environment in which Portland cement sets can be fully predicted. Determining the phase composition of the RS hydration slurry in alkaline solution would show whether there is any point at all in introducing it into the mix with OCP and whether it is reactive.
- What was the reason for introducing PG into the mix of OCP and RS. Calcium sulphate dihydrate is used as a setting time regulator for Portland clinker. The essence of the reaction is to inhibit the hydration of C3A in Portland clinker. In the OPC used, there is an additive that regulates the setting time and there are no phases in the RS that would need to be deactivated by the PG additive. What, then, was the purpose of the PG additive?
- Please add EDS analysis to the scanning microscope images. Highlighting the areas in the images does not allow us to check whether the authors have in fact well collated the morphology with the elemental composition of the micro-areas.
- Why did the authors of this paper not perform a quantitative XRD analysis of the OCP and RS mixture slurries? A quantitative analysis together with a thermal analysis could have shown whether there is any point in making mixtures with RS added. The XRD and DTA, TG, DTG results show that the changes in the intensity of the thermal effects, when confronted with the XRD results, demonstrate the lack of reactivity of the RS additive used and only a simple dilution effect of the OCP reactivity.
Since mdpi Materials is an international journal, all issues should be discussed based on the World's knowledge base. Therefore, the Introduction should include not only information about the occurrence of RS in China, but also in other places in the world (Europe, America Asia). Besides, attempts to replace OPC with RS have been made before, not only by scientists from China. There are many such positions in the literature databases which should be incorporated into an Introduction and Discussion parts.
Materials and Methods:
1. Please do not incorporate here measurements: content of lines 143-165 includes methods which are described later in the manuscript text. Please shift it to the beginning of Results section.
2. Please explain how Chinese standards can be compared to EU/ US ones
3. Line 172and Table 2 : explain W/B ratio (water to binder?)
4. Lines 219-221 contain discussion and not methods
5. Lines 222-226: Please give the purity and producer of KBr. Were spectra baselined/ corrected for air?
6. Please give the details concerning compression of tablets (line 223); press, producer, used pressure etc.
Several technical issues exist which should be corrected. Please find below some examples:
1. Please do not divide words in the title
2. Line 73 please add a full name since PG appears here for the first time (apart from Abstract)
3. Lines 97-98 is a repetition of line 43
4. Line 202: N/s
5. Line 228 to examine instead of to ascertain
6. References: Some names of journals are abbreviated some not, please check for the formatting of the references according to mdpi style
Figures:
1. Please correct description of axes on Figs 2,3,4, 8 and 12
2. Please baseline XRD spectrum of OPC on Fig.2
3. Fig 5. Add explanations to abbreviations on x axis. Flowability or Fluidity?
4. Fig 8. Enlarge peak annotations on Fig 8.
5. Fig 9. After 28 days of curation or samples cured for 28 days; the same throughout manuscript text
The language of manuscript is understandable from the context but it is suggested that it would be checked by a native speaker; some examples containing important information exclusively from Introduction as below:
Line 17 "excited volcanic ash activity of RS" should be "enhanced volcanic ash activity of RS" to convey the intended meaning.
Lines 21-22: "with a maximal reduction of 78.8% (RS-60)" should be "with a maximum reduction of 78.8% (RS-60)" for correct use of "maximum."
Line 22: "SiO2 active in the RS can react with Ca(OH)2 to produce C-S-H" should be "The active SiO2 in the RS can react with Ca(OH)2 to produce C-S-H" for improved clarity.
Line 24:"RS was responsible for dilution and infilling" should be "RS was responsible for dilution and filling" to use the appropriate term.
Lines 26-30: "This was primarily because the addition of PG provided the system with sufficient Ca2+ and SO42- to react with [Al(OH)6]3- to form ettringite (AFt), thereby accelerating the dissolution of Al3+ in RS to generate more AFt and C-(A)-S-H gels, which stimulated the volcanic ash activity of RS to some degree" is a lengthy and complex sentence.
Consider breaking it down into shorter, clearer sentences for better readability...etc
Author Response
Dear Reviewers:
We gratefully thank your constructive comments and helpful suggestions, which has significantly raised the quality of the manuscript and has enabled us to improve the manuscript. Each of your suggested revisions and comments was thoroughly considered and modified in the revised manuscript. we have added references to the Introduction section and resummarized the contents of the Introduction section. In the Materials and Methods section, we added the source of the FTIR test materials and sample preparation parameters and rearranged the text in this section. Revised and rearranged some figures. We have added references to support and discuss the SEM analysis in “Sections 3.4.4 and 3.5.4’’. We have also revised the entire article in English. As a result, the number and order of some figures, the order of the text, and the references in the text have been changed to reflect the changes in content. In this file, the authors will answer the comments item by item.Please see the attachment
Thank you again for your time and profound comments on our manuscript.
Best regards,
Chui-yuan Kong

Reviewer 3 Report
The presentation of the study is well-prepared. The following corrections need to be made for the study to be published:
· • In the "1.Introduction" section, RS, PG, and OPC should be written in their full form the first time they are mentioned.
· • In Table 2, although the code "RPO-5" is given, in some places like Fig.5, the code "PG-5" is used instead. This creates confusion. A single code should be used throughout the article.
· • There is not enough discussion in the "3.1.Flowability" section. This section should be supported by the literature. Additionally, it is not sufficiently stated why RS increases the flowability value.
· • In Fig.5, "PG-5" should be replaced with "RPO-5". Also, it is mentioned that the Flowability test was conducted three times and the average value was taken. Error bars should be provided on the graphs for these values.
· • Like in Fig.6 (a), the flexural strength values should also be provided in a graph format. It is stated that the compressive strength test was performed on 6 samples, while the flexural strength test was performed on 3 samples. Therefore, error bars should be provided on the graphs for these values.
· • In Fig.6 (a), it would be more appropriate to write the sample codes (RS-0, RS-15, RS-25, RS-35, RS-45, RS-50, and RS-60) instead of "Red sandstone ratio" on the "X-axis".
· • In temperature representations, there should be a space of 1 character between the "℃" symbol and the numbers.
· • The "3.3.4. SEM analysis" section is not sufficiently discussed with the literature. This section should be discussed with the literature.
· • The title "3.5. Effects of PG on the mechanical strength of RS-PG-OPC" may be written incorrectly. Because it is the same as the title "3.4. Effect of PG on the RS-PG-OPC's mechanical strength". Therefore, the titles need to be reviewed.
· • Suggestions for future studies in this area can be given in the "4. Conclusion" section.
· • The references need to be reviewed. Some journal names are written in their full forms, while others are written in abbreviated forms.
Author Response
Dear Reviewers:
We gratefully thank your constructive comments and helpful suggestions, which has significantly raised the quality of the manuscript and has enabled us to improve the manuscript. Each of your suggested revisions and comments was thoroughly considered and modified in the Revised Manuscript. In this file, the authors will answer the comments item by item.Please see the attachment.
Thank you again for your time and profound comments on our manuscript.
Best regards,
Chui-yuan Kong

Round 2
Reviewer 2 Report
The manuscript has been thoroughly revised. The Authors attached the detailed answer for my comments and explained some issues in detail.
The Figures have been greatly improved, some of the details were added (such as details how samples for FTIR were prepared) which allows other scientists to repeat the experiments. Now the quality of the manuscript is highly improved. It is ready to be published. The only concern is minor misspellings as dema45nd in line 44. Please chect the text carefully before final publication.
English is now fine but still some mistypings exist such as dema45nd in line 44. Please chect the text carefully
Author Response
Dear Reviewers:
We would like to thank you again for your suggestions and comments, which have greatly improved the quality of the manuscript and enabled us to improve it. We have followed your comments and we have carefully checked the text for English spelling issues. In the revised manuscript, we have changed "dema45nd" to "demand" in line 44 and "fluidity" to "flowability" in lines 114 and 547, and "Aft" to "AFt" in line 560. Please see the attachment for the revised version.
Thank you again for your time and profound comments on our manuscript.
Best regards,
Chui-yuan Kong

Reviewer 3 Report
The authors have made the necessary changes. Therefore the manuscript can be accepted.
Author Response
Dear Reviewers:
Thank you for taking time out of your busy schedule to review our manuscript and for your valuable comments during the review process, your comments and suggestions have greatly improved the quality of the manuscript. Thanks for your recognition of this study and your acceptance of our manuscript. Also, your comments and suggestions have been of great help for our future research.
Thank you again for accepting our manuscript and for your valuable comments and suggestions.
Best regards,
Chui-yuan Kong